# Egocentric Cross-Embodiment Video Editing via Dual Contrastive Representation Learning

## Abstract

Learning robotic manipulation from human videos is a promising solution to the data bottleneck in robotics, but the distribution shift between humans and robots remains a critical challenge. Existing approaches often produce entangled representations, where task-relevant information is coupled with human-specific kinematics, limiting their adaptability. We propose a generative framework for cross-embodiment video editing that directly addresses this by learning explicitly disentangled task and embodiment representations. Our method factorizes a demonstration video into two orthogonal latent spaces by enforcing a dual contrastive objective: it minimizes mutual information between the spaces to ensure independence while maximizing intra-space consistency to create stable representations. A parameter-efficient adapter injects these latent codes into a frozen video diffusion model, enabling the synthesis of a coherent robot execution video from a single human demonstration, without requiring paired cross-embodiment data. Experiments show our approach generates temporally consistent and morphologically accurate robot demonstrations, offering a scalable solution to leverage internet-scale human video for robot learning.

## 1 Introduction

The development of generalist embodied agents for unstructured real-world environments has been a longstanding goal in robotics (McCarthy et al., 2025). While foundation models in natural language processing (NLP) (Achiam et al., 2023; DeepSeek-AI, 2024) and computer vision (CV) (Betker et al., 2023; Labs et al., 2025; Wu et al., 2025; Yang et al., 2024) have succeeded by training on large, diverse internet datasets, robotics faces a persistent data bottleneck. Collecting large-scale robotic demonstration data requires substantial resources and complex procedures, which limits the scope and generalization of learned policies. This constraint has led researchers to explore human videos from the internet as a scalable data source (Hoque et al., 2025; Banerjee et al., 2024). These videos contain diverse tasks, physical interactions, and goal-oriented behaviors that could provide supervisory signals for training robotic skills.

However, learning from human video (LfV) presents a fundamental challenge that prevents direct application of standard imitation learning methods Chi et al. (2023); Liu et al. (2024b): a substantial distribution shift McCarthy et al. (2025) exists between human demonstrators and robotic learners. This "embodiment gap" encompasses visual differences in appearance (Lepert et al., 2025a) and viewpoint, as well as critical morphological and dynamic differences arising from distinct kinematics, degrees of freedom, and physical affordances between human hands and robotic end-effectors. This gap makes the direct transfer of skills from human video a formidable problem.

Existing approaches have attempted to bridge this gap with several strategies, yet they face key limitations. One line of research attempts to create intermediate representations to abstract away embodiment differences. This includes methods that define a unified state-action space to align human and robot data (Kareer et al.; Qiu et al., 2025) or those that learn a high-level plan from human video to guide a low-level robot controller (Zhu et al., 2024; Wang et al., 2023). However, they often result in entangled representations where task-relevant information remains coupled with human-specific kinematic biases. A second line of work employs visual adaptation to modify the demonstration

videos directly. These methods typically involve a procedural pipeline: inpainting removes the human agent, and a rendered robot model from a simulator is overlaid onto the scene (Lepert et al., 2025b;a). This approach is constrained by the photorealism of the simulator and, more critically, the simple overlay process can produce unrealistic video plans which are unsuitable for downstream policy learning.

Faced with these challenges, we aim to design a generative framework for cross-embodiment video editing that explicitly learns disentangled representations of task and embodiment. We hypothesize that a demonstration video can be decomposed into two orthogonal components: a latent representation of the task capturing the goal, object dynamics, and interaction sequence, and a separate representation of the embodiment capturing the agent's morphology and kinematics. Specifically, we introduce a framework built upon a large-scale, pre-trained generative model (VACE (Jiang et al., 2025)), which serves as a frozen generative backbone. Our approach augments this backbone with trainable encoders and a parameter-efficient adapter to factorize a source video into the distinct task and embodiment representations. Crucially, this disentanglement is enforced during training through a dual contrastive objective. This objective concurrently minimizes mutual information between the task and embodiment spaces using CLUB estimator (Cheng et al., 2020) to ensure their independence, while simultaneously maximizing intra-space consistency within each space via an InfoNCE (Oord et al., 2018) loss to create stable representations. At inference, the framework extracts the task representation from a human video, computes a representation for a target robot's embodiment, and composes them to synthesize a novel, robot-centric execution of the task. The resulting video provides a coherent, robot-specific visual demonstration, translating the human's actions into a dense visual plan. Our contributions are summarized as follows:

- We propose a generative, disentangled representation learning framework that addresses the key challenge of distribution shift in learning from human video.

- We augment a frozen video diffusion model with trainable encoders and a parameter-efficient adapter, trained via a dual contrastive objective that explicitly disentangles task and embodiment representations by minimizing mutual information between them while maximizing consistency within each representation space.

- Through extensive experiments, we demonstrate that our model synthesizes temporally coherent and morphologically consistent robot demonstration videos from a single human video, providing a scalable data source to help mitigate the data bottleneck in robotic imitation learning.

## 2 RELATED WORK

### 2.1 LEARNING FROM VIDEOS

Learning policies from human videos requires addressing the discrepancy between human and robot embodiments. One approach learns unified representation spaces where human and robot data can be aligned (Kareer et al.; Qiu et al., 2025). This alignment is achieved through shared policy backbones or embodiment-agnostic representations of interaction, such as joint motion manifolds (Park et al., 2025) or relative spatial relationships between agents and objects (Wei et al., 2024b). Visual domain adaptation provides an alternative strategy, editing human demonstration videos by inpainting humans and overlaying rendered robots (Lepert et al., 2025b; Li et al., 2025a). To address missing explicit action labels in video data, research has focused on learning intermediate representations. Rather than mapping directly from pixels to low-level robot controls, these methods translate visual demonstrations into abstract, transferable formats. Intermediate representations include latent action plans (Wang et al., 2023; Ye et al., 2024; Li et al., 2025b; Kim et al., 2025), sequences of end-effector pose affordances (Nasiriany et al., 2024), task-specific parameterizations such as screw motions for bimanual tasks (Bahety et al., 2024), and structured, object-centric state graphs (Zhu et al., 2024). Actions can also be inferred directly from visual information using geometric cues such as dense correspondences between video frames (Ko et al., 2023) or by matching task-invariant features between demonstrations and current scenes (Zhang & Boularias, 2024). Generative models have become important tools for scaling robot learning to large, uncurated video datasets. Video prediction models, pre-trained on internet and robot data, serve as implicit world models that learn physical dynamics and provide predictive visual representations for inverse dynamics control (Hu

et al., 2024; Wen et al., 2024). Generative models also enable large-scale data synthesis, creating datasets of synthetic trajectories for training generalizable policies (Jang et al., 2025). To utilize internet video data, retrieval-based frameworks use Vision Language Models (VLM) to identify relevant clips from unlabeled videos based on task descriptions (Papagiannis et al., 2024), while other systems convert internet videos into simulation environments for policy learning (Ye et al., 2025).

## 2.2 VIDEO MOTION CUSTOMIZATION

Recent advances in video generation emphasize motion customization, defined as preserving motion attributes from a source video, such as direction, speed, and pose, while modifying the dynamic object's appearance. One prominent strategy involves the disentanglement of motion from appearance, allowing for independent control. Initial work in this area includes models like MotionDirector (Zhao et al., 2024) and DreamVideo (Wei et al., 2024a). This concept was further refined by frameworks such as Motion Inversion (Wang et al., 2025a), which introduced a more explicit method using temporally coherent motion embeddings for customization. A more direct approach to motion control is through explicit spatiotemporal guidance. This can be achieved with trajectory-based methods, which evolved from early explorations like DragNUWA (Yin et al., 2023) to interactive, drag-based interfaces such as DragAnything (Wu et al., 2024). For more precise control, models like MotionPro (Zhang et al., 2025) utilize region-wise trajectory and motion masks to alleviate misinterpretation and improve accuracy. Pose-based guidance offers another powerful control mechanism, where generation is conditioned on skeletal sequences. Notable examples include Follow Your Pose (Ma et al., 2024), which employs a two-stage training strategy for high-quality character videos, and MimicMotion (Zhang et al., 2024), which achieves high-fidelity human motion synthesis through confidence-aware pose guidance. Furthermore, architectures like UniAnimate-DiT (Wang et al., 2025b) demonstrate the effective integration of motion tokens into large-scale video diffusion transformers for coherent animation. These control modalities are increasingly being integrated into more comprehensive, universal frameworks. Models like VACE (Jiang et al., 2025) are designed for "all-in-one" video creation and editing by accepting a diverse range of multimodal inputs. This research has also enabled highly specialized applications, such as the precise insertion of objects into existing scenes, a task effectively addressed by frameworks like VideoAnydoor (Tu et al., 2025).

## 2.3 EGOCENTRIC VIDEO DATASETS

Egocentric video research has evolved from foundational datasets such as "Something Something" (Goyal et al., 2017), which focuses on visual common sense through templated interactions, and BridgeData V2 (Walke et al., 2023), which addresses scalable robot learning through natural language processing. Recent work has produced large-scale human video corpora including Ego4D (Grauman et al., 2022), GigaHands (Fu et al., 2024), and EgoDex (Hoque et al., 2025). These datasets contain thousands of hours of diverse daily activities and dexterous manipulations, with multimodal annotations including 3D poses, gaze tracking, audio, and text descriptions. Specialized datasets such as HOT3D (Banerjee et al., 2024) and ARCTIC (Fan et al., 2023) provide high-precision 3D ground truth through multi-view and motion capture systems for detailed hand-object and articulated object manipulation analysis. In robotics, the Open X-Embodiment project (O'Neill et al., 2024) consolidates multiple datasets to support generalist policy development. These efforts advance understanding of complex interactions for embodied AI while addressing challenges in cross-embodiment generalization.

## 3 METHOD

In this section, we first describe our task formulation in Sec.3.1. Then we present the pipeline and the core of our proposed framework in Sec.3.2 and Sec.3.3. Sec.3.4 introduces our dataset construction. The overview of our framework is shown in Fig.1.

### 3.1 TASK FORMULATION

Our objective is to learn a generative model for cross-embodiment video editing, addressing the embodiment gap by learning to disentangle a video's content into two orthogonal latent representations. Given a demonstration video from a source agent with one embodiment (e.g., a human hand), we

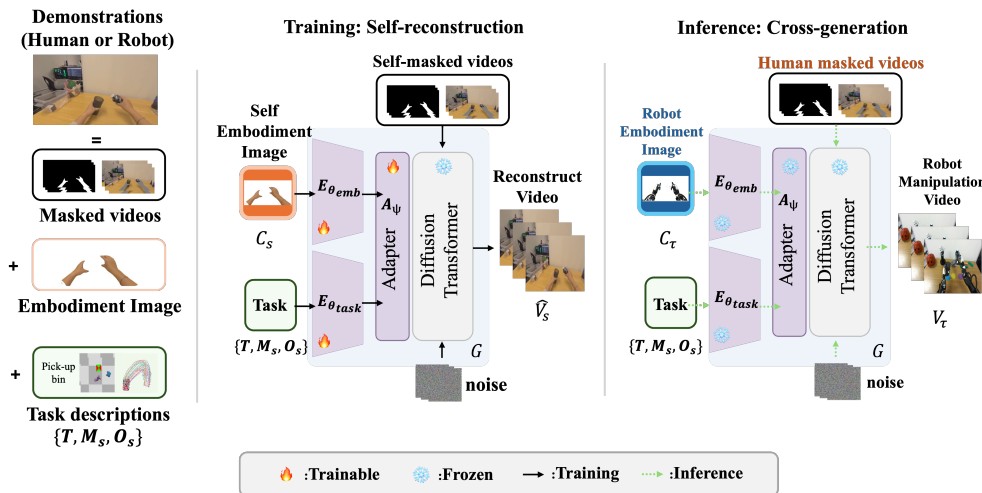

Figure 1: Framework Overview. Our framework for cross-embodiment video editing learns to disentangle a demonstration into two orthogonal latent representations. First, a trainable Task Encoder creates an embodiment-invariant Task Embedding from the text description, hand motion, and object trajectory. Simultaneously, an Embodiment Encoder generates an Embodiment Embedding from a static image of the agent's end-effector. This disentanglement is enforced by a dual contrastive objective that minimizes mutual information between the two spaces while maximizing consistency within each. The embeddings are injected into a frozen Diffusion Transformer via a parameter-efficient Adapter. The generative model is also conditioned on the static background to focus synthesis on the agent. In inference, the task embedding from a source human video can be composed with a target robot's embodiment embedding to synthesize a novel, robot-centric execution of the task.

synthesize a video depicting a target agent with different morphology (e.g., a robotic end-effector) executing the same task.

Formally, let $V_s = \{I_1^s, I_2^s, \ldots, I_N^s\}$ be a source video of $N$ frames showing a task performed by a source embodiment $e_s$. Let $C_\tau$ be the conditioning information that specifies the target embodiment $e_\tau$, such as one or more images of the robotic end-effector. The task is to generate a photorealistic and physically plausible target video $V_\tau = \{I_1^\tau, I_2^\tau, \ldots, I_N^\tau\}$ that shows embodiment $e_\tau$ performing the task demonstrated in $V_s$.

## 3.2 OVERVIEW AND TRAINING OBJECTIVES

Our approach learns a disentangled representation from the source video $V_s$ by factorizing it into two components: an embodiment-invariant task representation $z_{\text{task}}$ and an embodiment-specific representation $z_{\text{emb}}$. The task representation $z_{\text{task}}$ encodes the abstract semantics of the manipulation, such as the goal, object interactions, and motion, while $z_{\text{emb}}$ captures the agent's specific morphology and kinematics.

To instantiate this, we define the task representation $z_{\text{task}}$ as a multi-modal encoding extracted from the source video $V_s$. It is composed of a textual goal description $T$, the end-effector motion $M_s$ (a sequence of per-frame 3D positions, 6D rotations, and grip states), and the object trajectory $O_s$ (the sequence of 3D positions and 6D rotations for all manipulated objects).

A trainable task encoder $E_{\theta_{\text{task}}}$ maps these modalities to a latent embedding $z_{\text{task}} = E_{\theta_{\text{task}}}(T, M_s, O_s)$, while an embodiment encoder $E_{\theta_{\text{emb}}}$ maps a static end-effector image $C$ to its own embedding $z_{\text{emb}} = E_{\theta_{\text{emb}}}(C_s)$.

To avoid the need for paired cross-embodiment data, we train these encoders via a self-reconstruction objective. A generative model $G$, comprising a frozen video diffusion backbone and a trainable adapter $\mathcal{A}_\psi$, learns to reconstruct the source video $V_s$ from random noise $\epsilon$, conditioned on its own

disentangled embeddings:

$$\hat{V}_s = G(\epsilon, z_{\text{task}}, z_{\text{emb}}^s; \mathcal{A}_\psi) = G(\epsilon, E_{\theta_{\text{task}}}(T, M_s, O_s), E_{\theta_{\text{emb}}}(C_s); \mathcal{A}_\psi).$$

We optimize the trainable parameters $\theta$ of the encoders and adapter using a Flow Matching ($\mathcal{L}_{\text{FM}}$) objective based on Rectified Flows (Liu et al., 2022). This framework defines a linear interpolation path $x_t = tx_1 + (1 - t)x_0$ between a noise sample $x_0 \sim \mathcal{N}(0, I)$ and the target video latent $x_1$. This path has a constant ground truth velocity $v_t = x_1 - x_0$. Our model's velocity predictor $u_\theta$ is trained to estimate this vector, conditioned on the time $t$ and the embeddings, by minimizing the mean squared error:

$$\mathcal{L}_{\text{FM}} = \mathbb{E}_{x_0, x_1, t, z_{\text{task}}, z_{\text{emb}}} \left[ \|u_\theta(x_t, t, z_{\text{task}}, z_{\text{emb}}) - v_t\|^2 \right],$$

where $\theta = \{\theta_{\text{task}}, \theta_{\text{emb}}, \phi\}$.

At inference, our disentangled representations enable zero-shot skill transfer. Given a source human video $V_s$ and a target robot image $C_\tau$, we extract the invariant task representation $z_{\text{task}}$ from the human video and the target embodiment representation $z_{\text{emb}}^\tau$ from the robot image. The generator $G$ synthesizes a robot-specific video demonstration $V_\tau$ by composing the human's task with the robot's embodiment:

$$V_\tau = G(\epsilon, z_{\text{task}}, z_{\text{emb}}^\tau; \mathcal{A}_\psi).$$

The resulting video $V_\tau$ translates the human's actions into a dense visual plan for the target robot.

### 3.3 MODEL ARCHITECTURE

Our architecture integrates disentangled representations into a pre-trained video synthesis backbone. It consists of three main components: a frozen video diffusion model, trainable encoders for the multi-modal task and embodiment signals, and a lightweight adapter for parameter-efficient conditional generation.

**Frozen Video Generation Backbone.** Our generative backbone is the pre-trained VACE model (Jiang et al., 2025), whose parameters remain frozen during training. VACE is a latent Diffusion Transformer (DiT) designed for multimodal video synthesis. It operates by first encoding inputs into a latent space via a 3D Variational Autoencoder (VAE). A Video Condition Unit (VCU) then processes these multimodal latent signals to condition the diffusion process. By freezing VACE, we leverage its strong generative prior for synthesizing high-quality, temporally coherent video in a parameter-efficient manner.

**Trainable Conditioning Encoders.** The task and embodiment conditioning signals are processed by a set of trainable encoders into latent representations. Each encoder uses a `[CLS]` token (Devlin et al., 2019) architecture to produce a fixed-length embedding, where a prepended learnable token aggregates global information from the input sequence via self-attention.

The task encoder, $E_{\theta_{\text{task}}}$, processes the text prompt $T$, hand motion $M_s$, and object trajectory $O_s$. The text prompt is first encoded using the frozen VACE text encoder (Raffel et al., 2020), and its features are then passed to a shallow trainable Transformer to produce a text embedding. Similarly, the hand motion and object trajectory sequences are first projected into a feature space and then processed by separate Transformer encoders. The resulting text, motion, and trajectory embeddings are concatenated and fused by a multi-layer perceptron (MLP) to yield the final task embedding $z_{\text{task}}$.

For the embodiment encoder $E_{\theta_{\text{emb}}}$, we first extract patch-level features from the end-effector image using the frozen VACE CLIP (Radford et al., 2021) encoder. These features are then input to a shallow Transformer to produce the final embedding $z_{\text{emb}}$.

**Adapter for Conditional Control.** We inject the disentangled embeddings into the frozen DiT backbone using a lightweight, trainable adapter $\mathcal{A}_\psi$ inspired by the Context Adapter Tuning strategy (Jiang et al., 2025). The adapter consists of a series of Transformer blocks that mirror the architecture of the frozen backbone. It processes the concatenated task and embodiment embeddings ($z_{\text{task}}, z_{\text{emb}}$) in a parallel stream. The output features from each adapter block are then added element-wise to the features of the corresponding block in the frozen backbone. This additive injection allows the task and embodiment signals to steer the video generation at multiple feature levels without modifying the original model weights, enabling parameter-efficient control.

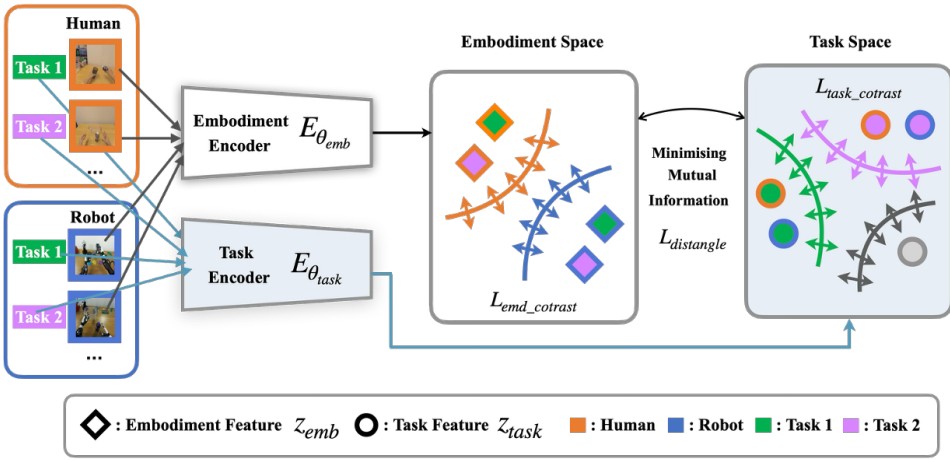

Figure 2: The Dual Contrastive Learning Objective. Our objective structures the latent space by simultaneously minimizing mutual information between the task and embodiment spaces ($L_{\text{distangle}}$) while maximizing intra-embodiment-space consistency ($L_{\text{emb\_contrast}}$) and maximizing intra-task-space consistency ($L_{\text{task\_contrast}}$).

### 3.4 CONTRASTIVE LEARNING FOR DISENTANGLED REPRESENTATIONS

To ensure the learned task embedding $z_{\text{task}}$ and embodiment embedding $z_{\text{emb}}$ are disentangled, we introduce a dual contrastive learning objective (Fig. 2). This objective structures the latent space by concurrently minimizing the mutual information between task and embodiment representations to promote their independence, while simultaneously maximizing the mutual information between embeddings of the same semantic class (i.e., same task or same embodiment) to ensure the representations are robust and discriminative.

**Minimizing Cross-Representation Mutual Information.** We enforce independence between $z_{\text{task}}$ and $z_{\text{emb}}$ by minimizing their mutual information (MI), $I(z_{\text{task}}; z_{\text{emb}})$. To this end, we employ the CLUB estimator (Cheng et al., 2020), which provides an upper bound on MI suitable for minimization. We train a variational model $q_\phi(z_{\text{emb}}|z_{\text{task}})$ [1] to approximate the conditional distribution $p(z_{\text{emb}}|z_{\text{task}})$. The disentanglement loss, $\mathcal{L}_{\text{disentangle}}$, contrasts the log-likelihood of naturally paired samples against that of randomly paired samples:

$$\mathcal{L}_{\text{disentangle}} = \mathbb{E}_{p(z_{\text{task}}, z_{\text{emb}})}[\log q_\phi(z_{\text{emb}}|z_{\text{task}})] - \mathbb{E}_{p(z_{\text{task}})p(z_{\text{emb}})}[\log q_\phi(z_{\text{emb}}|z_{\text{task}})]. \quad (1)$$

Minimizing this loss drives $q_\phi(z_{\text{emb}}|z_{\text{task}})$ to become independent of $z_{\text{task}}$, thereby minimizing the MI and promoting disentanglement.

**Maximizing Intra-Representation Similarity.** To structure each latent space, we apply a contrastive loss that pulls embeddings of the same class together while pushing dissimilar ones apart. We use the InfoNCE (Oord et al., 2018) loss for both task and embodiment representations. For an anchor embedding $z_i$, a positive sample $z_i^+$ from the same class, and a set of negative samples $\{z_k^-\}$ from different classes, the loss is:

$$\mathcal{L}_{\text{contrast}} = -\mathbb{E}\left[\log \frac{\exp(\text{sim}(z_i, z_i^+))}{\exp(\text{sim}(z_i, z_i^+)) + \sum_k \exp(\text{sim}(z_i, z_k^-))}\right], \quad (2)$$

where $\text{sim}(\cdot, \cdot)$ is cosine similarity.

To enforce task invariance, we apply this loss as $\mathcal{L}_{\text{task\_contrast}}$, where positive pairs are task embeddings ($z_{\text{task}}$) from different videos depicting the same task, and negatives are from videos of different tasks. Similarly, to enforce embodiment consistency, we apply the loss as $\mathcal{L}_{\text{emb\_contrast}}$, where positive pairs are embodiment embeddings ($z_{\text{emb}}$) from different images of the same agent, and negatives are from images of different agents.

---

[1]detail see Appendix

The final training objective is a weighted sum of the Flow Matching reconstruction loss and the auxiliary contrastive losses:

$$\mathcal{L} = \mathcal{L}_{\text{FM}} + \lambda_{\text{dis}}\mathcal{L}_{\text{disentangle}} + \lambda_{\text{task}}\mathcal{L}_{\text{task\_contrast}} + \lambda_{\text{emb}}\mathcal{L}_{\text{emb\_contrast}}. \tag{3}$$

The primary $\mathcal{L}_{\text{FM}}$ term ensures reconstruction fidelity, while the auxiliary contrastive losses regularize the latent space to enforce disentanglement and intra-space similarity. The $\lambda$ hyperparameters balance the influence of these objectives during optimization.

### 3.5 DATASET PREPARATION

Our training data is derived from the Physical Human-Humanoid Data (PH$^2$D) dataset (Qiu et al., 2025), a task-oriented subset of EgoDex (Hoque et al., 2025). The dataset provides synchronized egocentric video with high-fidelity human motion data, but lacks explicit 6D poses for manipulated objects. To address this, we developed an automated pipeline to extract these object trajectories.

Agent hand motion is sourced directly from the high-quality SE(3) pose annotations provided in PH$^2$D. The object trajectories, however, require a multi-stage extraction pipeline. First, we perform 2D object tracking using Grounding DINO (Liu et al., 2024a) for initial detection and SAM2 (Ravi et al., 2024) for precise mask-based tracking. Next, we lift these 2D tracks to 3D by generating a dense depth map for each frame with the Video Depth Anything model (Chen et al., 2025), allowing us to construct a per-frame 3D point cloud of the object. Finally, we estimate the full SE(3) pose trajectory by applying the Iterative Closest Point (ICP) algorithm (Chetverikov et al., 2002) between consecutive point clouds to recover the object's rigid body motion. The final prepared dataset consists of egocentric video sequences, each paired with temporally aligned SE(3) pose trajectories for both the agent's hands and the primary manipulated object.

We use SAM2 to segment and track the human (robot) hand, followed by a morphological dilation kernel to expand the mask slightly, ensuring the entire human limb is covered.

## 4 EXPERIMENTS

### 4.1 EXPERIMENTAL SETUP

**Implementation Details.** We employ the pre-trained Wan2.1-VACE-1.3B Jiang et al. (2025) as the generative backbone. Our framework introduces three trainable components: a task encoder, an embodiment encoder, and a parameter-efficient adapter. Both encoders are lightweight Transformers that map inputs to fixed-length embeddings via a `[CLS]` token and a single self-attention layer with 64 hidden dimensions. The adapter consists of 15 Transformer blocks, initialized with the corresponding pre-trained layers of the frozen VACE backbone to accelerate convergence. Training is conducted for two epochs on 8 NVIDIA H200 GPUs with a global batch size of 40, using AdamW Loshchilov & Hutter (2017) with a learning rate of $1 \times 10^{-5}$. To estimate conditional mutual information, we adopt the CLUB estimator, which trains a separate variational model to approximate $p(z_{\text{emb}} \mid z_{\text{task}})$ via maximum likelihood, optimized with a learning rate of $1 \times 10^{-4}$. For stability, gradients from the disentanglement loss $\mathcal{L}_{\text{disentangle}}$ are applied to the main model only once every ten training steps. The generated video runs at 15 frames per second, and we utilize 81 video frames (5 seconds) for training. At inference, we generate videos using 50 denoising steps with a VACE scale of 1.0. The loss weights are set to $\lambda_{\text{dis}} = 1.0$, $\lambda_{\text{task}} = 0.5$, and $\lambda_{\text{emb}} = 0.5$.

**Baselines.** We compare our framework against the following baseline methods. (1) **VACE** (Jiang et al., 2025): The general-purpose video editing model, which we task with inpainting the robot by providing the masked video, the corresponding mask, a target robot image, and a text description as conditional inputs. (2) **Phantom** (Lepert et al., 2025b): A procedural method that first removes the human hand via video inpainting and then overlays a rendered robot model animated with the extracted human motion. We reproduce these baseline methods using their officially released code and recommended configurations for a fair comparison.

**Metrics.** We first employ metrics of PSNR, SSIM Wang et al. (2004), LPIPS Zhang et al. (2018), and FVD Unterthiner et al. (2018) to evaluate the video fidelity of the generated video. To further evaluate video quality and consistency, we utilize a selection of metrics from VBench Huang et al.

Table 1: Quantitative comparison with baseline methods on the cross-embodiment video editing task. Our method demonstrates superior performance across both video fidelity and a comprehensive set of VBench quality metrics. We report Aesthetic Quality (AQ), Imaging Quality (IQ), Background Consistency (BC), Subject Consistency (SC), Temporal Style (TS), Motion Smoothness (MS), Overall Consistency (OC), and Temporal Flickering (TF). Arrows (↑/↓) indicate whether higher or lower values are better.

| | Video Fidelity | | | | VBench Quality & Consistency | | | | | | | |
| Method | FVD (↓) | LPIPS (↓) | PSNR (↑) | SSIM (↑) | AQ (↑) | IQ (↑) | BC (↑) | SC (↑) | TS (↑) | MS (↑) | OC (↑) | TF (↑) |
|---|---|---|---|---|---|---|---|---|---|---|---|---|
| VACE (Jiang et al., 2025) | 1575.8 | 0.676 | 12.43 | 0.515 | 0.449 | **0.695** | **0.920** | **0.905** | 0.121 | 0.993 | 0.121 | 0.986 |
| Phantom (Lepert et al., 2025b) | 1948.3 | 0.676 | 10.96 | 0.435 | 0.501 | 0.624 | 0.886 | 0.812 | 0.122 | 0.962 | 0.122 | 0.940 |
| **Ours** | **1469.6** | **0.674** | **13.24** | **0.532** | **0.509** | 0.645 | 0.910 | 0.891 | **0.132** | **0.994** | **0.132** | **0.990** |

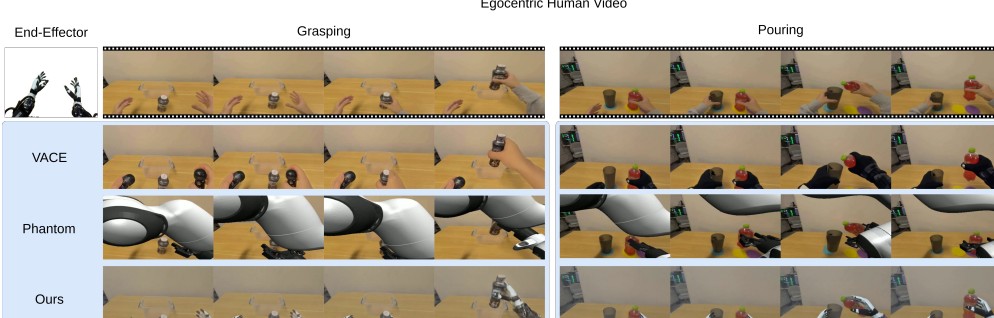

Figure 3: Qualitative comparison of cross-embodiment video editing. Given a source egocentric human video (top row) and a target robot end-effector (top left), we compare the synthesized videos from our method against the VACE and Phantom baselines across two manipulation tasks: Grasping and Pouring. The general-purpose VACE model struggles to generate the correct morphology, often inpainting a generic humanoid hand from its pre-training data. The procedural Phantom baseline produces an artificial-looking overlay that lacks realistic lighting and scene integration. In contrast, our method generates temporally coherent videos where the robot embodiment is morphologically correct, physically plausible, and well-integrated with the scene's dynamics.

(2024), which includes eight key indicators: aesthetic quality (AQ), imaging quality (IQ), background consistency (BC), subject consistency (SC), temporal style (TS), motion smoothness (MS), overall consistency (OC), and temporal flickering (TF).

## 4.2 COMPARISONS TO BASELINE METHODS

**Quantitative results.** Table 1 shows the quantitative results on the cross-embodiment video editing task. Our method consistently achieves the best performance across all video fidelity metrics, improving the FVD from 1575.8 (VACE) to 1469.6, and substantially outperforming the Phantom baseline (1948.3). Similar gains in LPIPS, PSNR, and SSIM indicate superior overall realism and reconstruction accuracy. The VBench evaluation reveals a more nuanced comparison; the general-purpose VACE model performs best on static-centric metrics like IQ, BC, and SC, leveraging its strong inpainting priors to maintain scene integrity. In contrast, our framework excels in metrics critical for dynamic and temporal quality, achieving the highest scores for AQ, TS, MS, and OC. This result indicates that by explicitly learning a disentangled representation, our model is more effective at generating temporally coherent motion for a new morphology, whereas general-purpose methods struggle with dynamic plausibility, and procedural overlays suffer from poor integration.

**Qualitative results.** We present a qualitative comparison in Fig. 3 on Grasping and Pouring tasks. Both baselines exhibit significant failure modes. The VACE model fails to generate the correct robot morphology, hallucinating a humanoid hand. The procedural Phantom baseline, while displaying the correct shape, produces a non-photorealistic overlay that is poorly integrated with the scene. In contrast, our framework generates videos that are both morphologically accurate and temporally coherent. By disentangling the invariant task representation from the source video and composing it

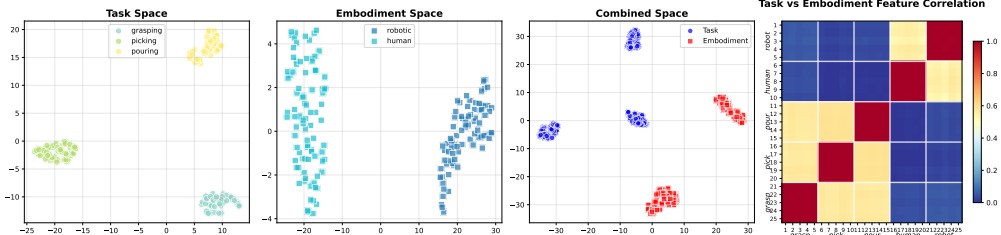

Figure 4: **Visualization of the disentangled latent spaces learned with our dual contrastive objective.** The t-SNE plots (left three) show clear clustering of different tasks and embodiments into distinct groups. The correlation matrix (right) quantitatively confirms the disentanglement, showing high intra-space similarity (red diagonal blocks) and near-zero correlation between task and embodiment representations (blue off-diagonal blocks), indicating that the two spaces are successfully disentangled.

with the target embodiment, our model synthesizes a plausible, robot-centric execution, forming a high-fidelity visual plan. We also provide additional results in the Appendix for further reference.

Figure 4 demonstrates that our dual contrastive objective successfully learns disentangled representations. The t-SNE visualizations show that embeddings for different tasks (grasping, picking, pouring) and embodiments (robotic, human) form distinct clusters. This separation is quantified by the feature correlation heatmap, which reveals high intra-class correlation (diagonal blocks) but near-zero correlation between the task and embodiment spaces (off-diagonal blocks). Together, these results confirm that the objective yields representations that are both internally consistent and mutually independent.

### 4.3 ABLATION STUDY

To validate the effectiveness of our dual contrastive objective, we conduct an ablation study on a zero-shot[2] "Picking" task, with results presented in Table 2. Removing the objective leads to a degradation in generation quality, evidenced by weaker performance across all fidelity metrics. This confirms that explicitly regularizing the latent space to enforce disentanglement between task and embodiment is critical for synthesizing high-fidelity videos. We also provide visual results in the Appendix for further reference.

Table 2: Ablation study on the dual contrastive objective of our framework.

| Method | FVD ($\downarrow$) | LPIPS ($\downarrow$) | PSNR ($\uparrow$) | SSIM ($\uparrow$) |
|---|---|---|---|---|
| w/o Dual Contrastive | 1557.8 | 0.635 | 13.42 | 0.489 |
| w/ Dual Contrastive | **1514.9** | **0.576** | **13.53** | **0.498** |

To validate the specific contributions of our proposed training objectives, we conducted a systematic ablation study by training variants of our model: without the disentanglement objective ($\mathcal{L}_{\text{disentangle}}$), without the intra-space contrastive objective ($\mathcal{L}_{\text{contrast}}$), and without the entire dual objective. The latter variant serves as a direct "VACE-Finetuned" baseline, utilizing only the flow-matching reconstruction loss. Analyzing the t-SNE visualizations of the learned latent spaces (provided in the Appendix), we observe that the full Dual Contrastive objective is essential for structuring the latent space. Removing the dual objective leads to an unstructured space where task and embodiment features overlap. Specifically, the VACE-Finetuned baseline fails to bridge the domain shift, often replacing the hand with black regions or retaining human morphology. Furthermore, removing $\mathcal{L}_{\text{disentangle}}$ reduces the separation between task and embodiment representations, while omitting $\mathcal{L}_{\text{contrast}}$ results in unclear task and embodiment clusters. Our full model, conversely, yields distinct and compact clusters, enabling the synthesis of morphologically accurate and temporally coherent robot demonstrations across diverse tasks.

---

[2]data not seen during training

## 4.4 DOWNSTREAM ROBOTICS VALIDATION

To demonstrate that our generated videos are not only visually realistic but also physically plausible for downstream control, we conducted a Visual Behavioral Cloning (BC) experiment to validate the transfer capability. We utilize the MuJoCo physics simulator with a high-DoF Unitree H1 humanoid robot to perform a precise manipulation task ("Grasping Pepsi").

For the policy architecture, we employ ACT Zhao et al. (2023) with a ResNet backbone. We compare our approach against HAT Qiu et al. (2025), a state-of-the-art baseline that aligns human and robot data for policy learning. For our method, we first translate the human demonstration videos into the target robot's visual domain using our proposed video editing framework. We then train the policy using these generated robot videos.

Table 3: Policy precision on the downstream Unitree H1 grasping task. We report the validation loss on held-out robot trajectories, including total loss, Action L1 error, and End-Effector (EEF) position loss.

| Method | Total Val Loss ($\downarrow$) | Action L1 Error ($\downarrow$) | EEF Pos Loss ($\downarrow$) |
|---|---|---|---|
| HAT (Qiu et al., 2025) | 0.2006 | 0.019 | 0.091 |
| **Ours** | **0.1886** | **0.018** | **0.085** |

The quantitative results are presented in Table 3. The policy trained on our generated videos achieves consistently lower validation errors across all metrics compared to the HAT baseline. This indicates that our method effectively bridges the cross-embodiment gap *before* policy training, providing kinematic supervision that is more aligned with the robot's physical embodiment than direct alignment methods. We provide video visualizations of the successful robot rollouts (in both egocentric and third-person views) and the generated training data in the supplementary material[3].

## 5 CONCLUSION

In this work, we addressed the key challenge of the distribution shift in learning from human video, where prior methods are often hindered by entangled representations that couple task semantics with human-specific kinematics. We introduced a generative framework for cross-embodiment video editing that directly tackles this by learning explicitly disentangled representations of task and embodiment. Our method successfully factorizes a demonstration video into two orthogonal latent spaces using a dual contrastive objective: a CLUB estimator minimizes mutual information to ensure independence, while an InfoNCE loss enforces intra-space consistency. A parameter-efficient adapter injects these representations into a frozen video generation backbone, enabling the synthesis of temporally coherent and morphologically accurate robot execution videos from a single human demonstration, without requiring paired cross-embodiment data. This approach offers a scalable solution to the data bottleneck in robotics, effectively translating human skills into actionable visual plans for diverse robot morphologies.

## ETHICS STATEMENT

This work utilizes publicly available egocentric video datasets ($PH^2D$). Our research did not involve the collection of new data from human subjects. We acknowledge that the datasets used may contain inherent biases (e.g., demographic, environmental, or task-related), which could be reflected in our trained model and downstream robotic policies. Addressing and mitigating these biases is an important direction for future research.

## REPRODUCIBILITY STATEMENT

To ensure the reproducibility of our results, we provide an anonymous downloadable source code at: `https://anonymous.4open.science/r/EgoXEdit-E19F`, and commit to making all

---

[3]Visual results available at: `https://anonymous.4open.science/r/Ego_re-3145`

resources publicly available upon publication. We also release the trained weights for our adapter and conditioning encoders. Our data processing pipeline, used to extract SE(3) object trajectories from the PH$^2$D dataset, is described in detail in Section 3.5. The baseline methods used for comparison were implemented using their officially released codebases, and all evaluation metrics are standard and well-established in the literature.

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

## A CLUB Variational Model Details

To implement the mutual information minimization via the CLUB estimator, we employ a variational approximation $q_\phi(z_{\text{emb}}|z_{\text{task}})$ parameterized by a Multi-Layer Perceptron (MLP). This network consists of three linear layers with GELU activations (and a Tanh output for the log-variance), mapping the task embedding $z_{\text{task}}$ to the mean $\mu$ and log-variance $\log \sigma^2$ of the conditional distribution. The optimization process is performed jointly: we alternate between updating the variational parameters $\phi$ to maximize the log-likelihood of true pairs $(z_{\text{task}}, z_{\text{emb}})$—thereby ensuring the approximation is accurate and the upper bound is valid—and updating the encoder parameters to minimize the estimated mutual information.

## B More experimental results

We provide additional qualitative results beyond the examples shown in Fig. 3 of the main paper.

## C Use of Large Language Models (LLMs)

We used large language models only as a light writing aid to improve grammar, phrasing, and formatting. The models were *not* used to propose ideas, design algorithms, write code, run experiments, analyze data, or draft scientific content. All technical claims, methods, and conclusions were conceived, executed, and verified by the authors. Any suggested edits were manually reviewed and incorporated at our discretion. In line with ICLR policy, we disclose this use and accept full responsibility for the accuracy and integrity of the manuscript, including ensuring that no plagiarized or misrepresented content produced by LLMs is included.

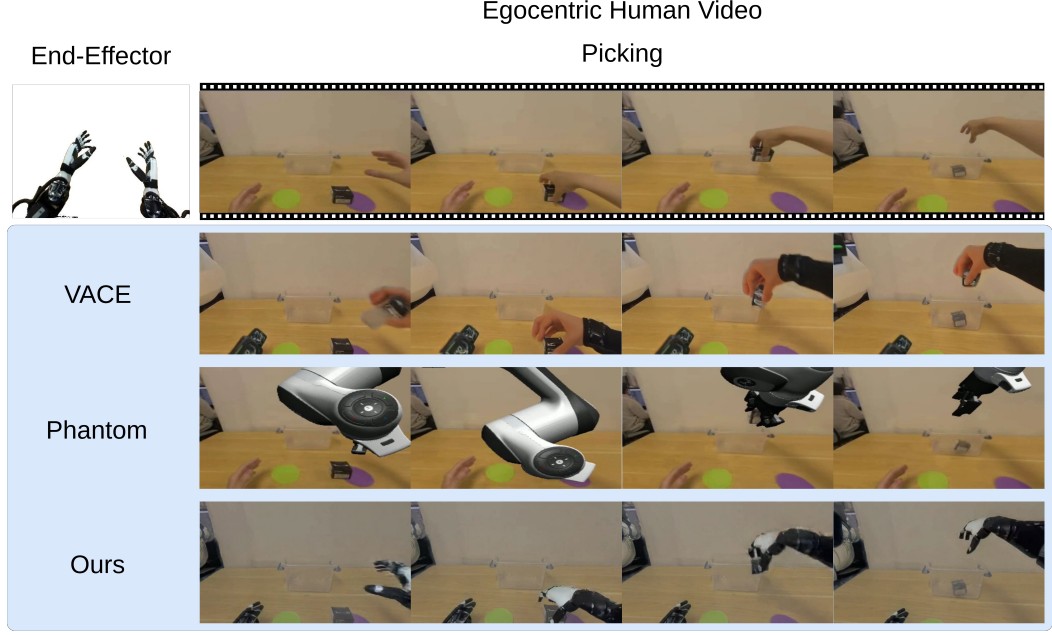

Figure 5: Qualitative comparison of cross-embodiment video editing on the 'grasping a plastic bottle' task.

Figure 6: Qualitative comparison of cross-embodiment video editing on the 'picking up a black box' task.

Egocentric Human Video

End-Effector                                      Pouring

VACE

Phantom

Ours

Figure 7: Qualitative comparison of cross-embodiment video editing on the 'pouring' task.

Egocentric Human Video

End-Effector                                      Grasping

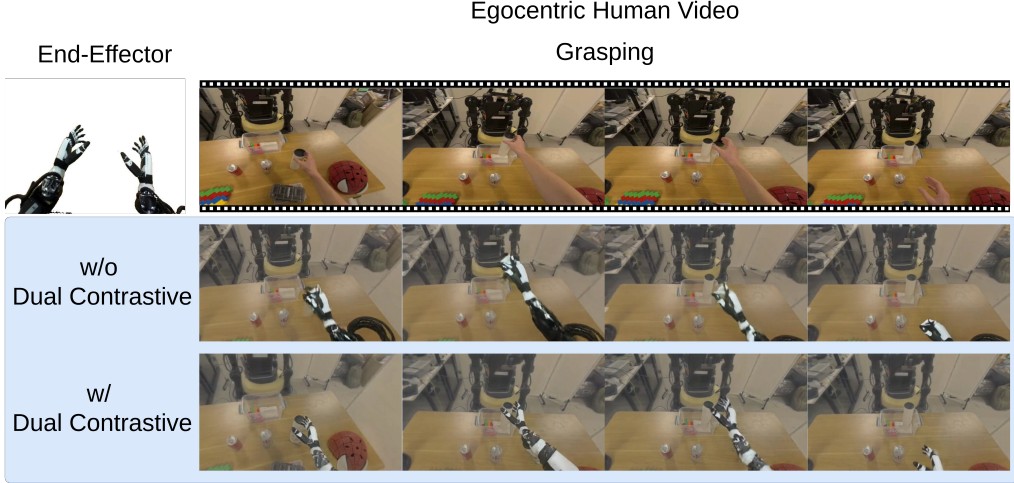

w/o
Dual Contrastive

w/
Dual Contrastive

Figure 8: Visual ablation study of the dual contrastive objective on the 'grasping' task. Removing the dual contrastive objective leads to a clear degradation in generation quality, evidenced by the blurred interaction between the robot hand and the object. Our full model produces a cleaner and more coherent result, confirming that our explicit latent space regularization is critical for synthesizing high-fidelity interactions.

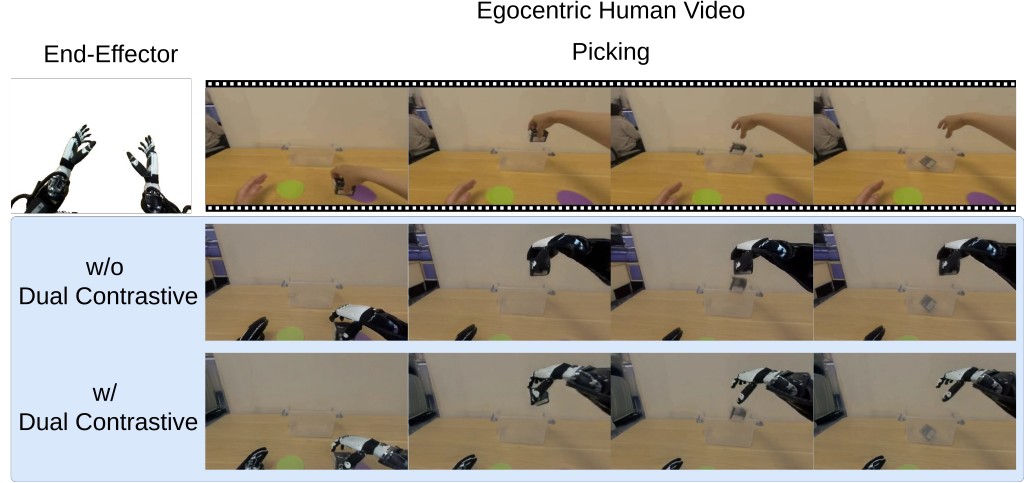

Figure 9: Visual ablation study of the dual contrastive objective on the 'picking' task.

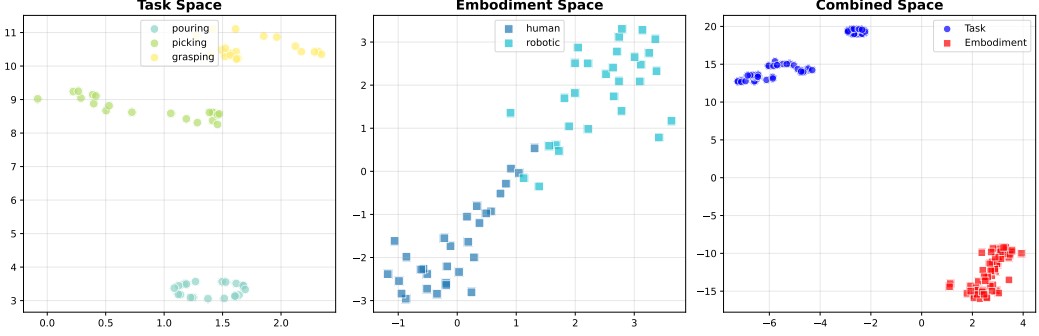

Figure 10: T-SNE ablation study of the disentangle objective.

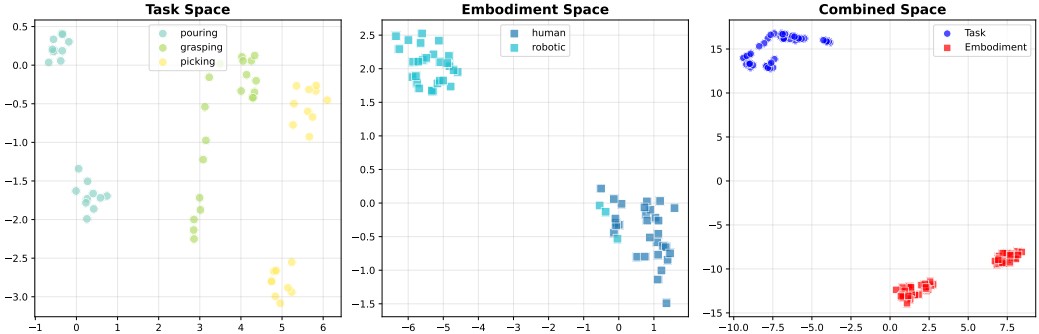

Figure 11: T-SNE ablation study of the task contrast objective.

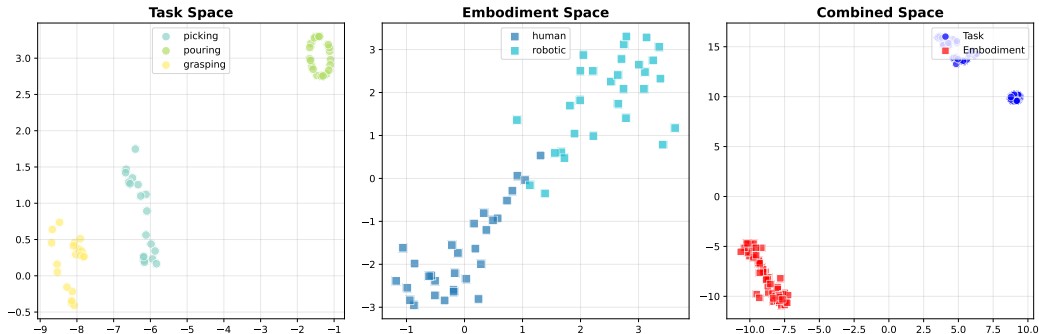

Figure 12: T-SNE ablation study of the embodiment contrast objective.

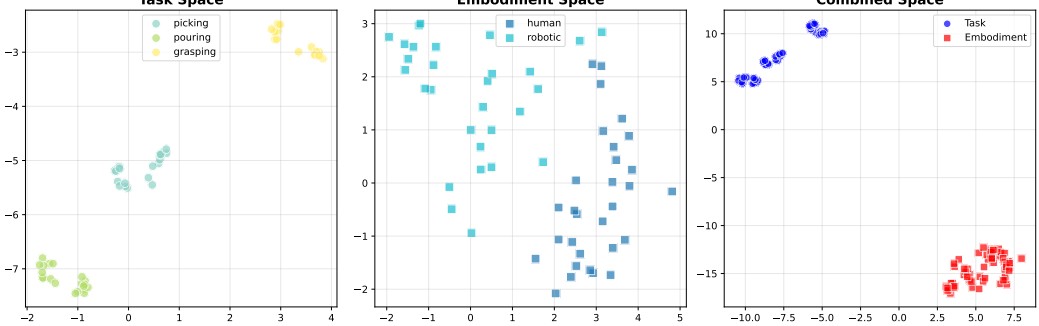

Figure 13: T-SNE ablation study of the dual objective.

