# OpenReview forum: "Egocentric Cross-Embodiment Video Editing via Dual Contrastive Representation Learning"
_ICLR.cc/2026/Conference — Submitted to ICLR 2026_

### Official Review · Reviewer_8Qp5 · 2025-10-25

**Soundness:** 3
**Presentation:** 3
**Contribution:** 2
**Rating:** 2
**Confidence:** 4

**Summary:**

This paper introduces a cross-embodiment video generation method. The main problem of using human videos for robot learning is the embodiment gap. This paper resolves the visual embodiment gap between human and robot by adopting a video generation approach. To disentangle task-centric information from embodiment-centric information, the authors propose a dual contrastive learning approach. Through the proposed objective function, a trainable adapter is trained while maintaining the ability of the frozen video generation backbone. Its generation quality outperforms previous baselines.

**Strengths:**

- The proposed objective function for dual contrastive learning, which incorporates the concept of disentangling task and embodiment information, is promising.

- The paper is well written, and the provided figures are helpful for understanding.

**Weaknesses:**

- **Insufficient explanation**
  - The paper provides limited information about the proposed method. For the masked videos, it is unclear what kind of method is used to create the masks. The training datasets used do not provide masks for each hand or robot end-effector. Therefore, the authors might have used an off-the-shelf method, which should be specified.
  - For the embodiment image, especially during inference, the authors do not state which embodiment image is used. Based on the provided figure, it seems to have a similar pose between the human and robot.
- **Limited analysis**
  - The main contribution of the paper is proposing a new objective function to disentangle task and embodiment information. However, the subsequent ablation study is not thoroughly executed. It only examines the removal of the entire objective function.
  - The authors should conduct an ablation study by removing each component of the objective function individually. For example, an ablation study on the mutual information loss could demonstrate the effectiveness of disentangling task and embodiment information. However, this paper lacks careful analysis of each component, even though it forms the core of the proposed method.
- **No robot experiment**
  - The main motivation of the paper is reducing the embodiment gap between human and robot for scalable robot training. However, the paper does not contain any robot downstream tasks.
  - The major distinction between computer vision and robotics lies in their application focus. The visual gap between humans and robots is a major bottleneck for robot learning, but various qualitative metrics such as FVD or LPIPS cannot guarantee effectiveness in real-world robot downstream tasks. For example, Phantom[1] demonstrates its effectiveness in reducing the embodiment gap by performing real-world robot tasks.
  - If the authors believe this paper falls under the category of generative methods, they should tone down the emphasis on robot learning and include further analysis regarding generative quality.
- **Missing citations**
  - H2R[2] is also a human-to-robot augmentation approach. It uses states such as hand poses and inpainted human video to generate robot datasets from human datasets.
  - UniSkill[3] is another learning from human approach. Its training objective ensures the extraction of embodiment-agnostic but task-centric latent skills from human videos. It also demonstrates cross-embodiment generation results from human to robot.
---
[1] Lepert, Marion, Jiaying Fang, and Jeannette Bohg. "Phantom: Training robots without robots using only human videos." arXiv preprint arXiv:2503.00779 (2025).

[2] Li, Guangrun, et al. "H2R: A Human-to-Robot Data Augmentation for Robot Pre-training from Videos." arXiv preprint arXiv:2505.11920 (2025).

[3] Kim, Hanjung, et al. "UniSkill: Imitating Human Videos via Cross-Embodiment Skill Representations." arXiv preprint arXiv:2505.08787 (2025).

**Questions:**

- How do you process the masked videos?
- For the self-embodiment image during inference, can it be any kind of robot embodiment image, or should it have a similar pose to the human?
- What's the effect of each objective function?
- What's the benefit of using a generative approach from human to robot? For the robot learning, rendering approaches like Phantom or H2R are sufficient for scalable robot learning.

---

> ### Author Response · Authors · 2025-11-29
> **Response to Reviewer 8Qp5**
>
> We sincerely thank the reviewer for the constructive feedback and for recognizing our dual contrastive objective as "promising." We agree that downstream validation and detailed ablations are critical for a robotics-focused paper. We have addressed your concerns through new experiments detailed in our **General Response**.
>
> **[W1 & Q1 & Q2] Method Clarifications**
>
> **Our Response:**
> * **Mask Generation:** We use **SAM2** [1] to segment and track the human/robot hand, followed by a morphological dilation kernel to expand the mask slightly, ensuring the entire human limb is covered.
> * **Embodiment Image at Inference:** The embodiment image does **not** need to have a similar pose to the human. Because our framework explicitly disentangles **motion (Task)** from **appearance (Embodiment)**, the Embodiment Encoder only needs a static image to extract the robot's morphology. The motion is driven entirely by the task representation extracted from the source video. We have included the specific conditioning images used in our experiments at the anonymous link.
>
> **[W2 & Q3] Detailed Ablation Study**
>
> **Our Response:** We acknowledge that our initial ablation was limited. We have now performed the component-wise ablation you requested (see **General Response** Section 2):
> *   **w/o Dual Objective:** Removing the entire Dual Contrastive Objective relies solely on the reconstruction loss ($\mathcal{L}_{FM}$), leading to a lack of overall alignment.
> *   **w/o Disentangle:** The reduced separation between representations in the absence of $\mathcal{L}_{disentangle}$ results from the removal of the mutual information minimization constraint.
> *   **w/o Contrast:** Removing the intra-space contrastive objectives causes structural degradation in their respective domains: omitting $\mathcal{L}_{contrast}$ leads to undefined and overlapping clusters in the Space.
>
>
> **[W3 & Q4] Downstream Robotics Validation & Generative vs. Rendering**
>
> **Our Response:** We agree that visual metrics are insufficient to guarantee real-world effectiveness.
> 1.  **New Robotics Experiment:** As detailed in our **General Response**, we conducted a **Visual Behavioral Cloning** [2] experiment using the **Unitree H1** robot in the MuJoCo simulator. We compared a policy trained on videos generated by our model against the **HAT** [3] baseline. Qualitative and quantitative results (see anonymous link) validate that our generative approach preserves the necessary kinematic structure for control, addressing your concern about the "application focus."
> 2.  **Generative vs. Rendering:** We respectfully argue that the fundamental benefit of our generative approach lies in its ability to learn **explicitly disentangled representations** via the dual contrastive objective, offering capabilities beyond the geometric transformations of rendering-based methods. While procedural approaches like Phantom can overlay robots, they often result in non-photorealistic artifacts with poor scene integration and do not extract transferable semantics. In contrast, our framework learns **embodiment-invariant task representations**. Furthermore, unlike fixed procedural pipelines, our learned representations are inherently generalizable, allowing the composition of extracted task semantics with arbitrary target embodiments to create physically plausible, photorealistic training data.
>
> **[W4] Missing Citations**
>
> **Our Response:** We thank you for pointing out these relevant works. We have added citations for **H2R** and **UniSkill** to our related work section.
>
> [1] Ravi, Nikhila, Valentin Gabeur, Yuan-Ting Hu, Ronghang Hu, Chaitanya Ryali, Tengyu Ma, Haitham Khedr et al. "Sam 2: Segment anything in images and videos." arXiv preprint arXiv:2408.00714 (2024).
>
> [2] Zhao, Tony Z., Vikash Kumar, Sergey Levine, and Chelsea Finn. "Learning fine-grained bimanual manipulation with low-cost hardware." arXiv preprint arXiv:2304.13705 (2023).
>
> [3] Qiu, Ri-Zhao, Shiqi Yang, Xuxin Cheng, Chaitanya Chawla, Jialong Li, Tairan He, Ge Yan et al. "Humanoid Policy~ Human Policy." arXiv preprint arXiv:2503.13441 (2025).

---

### Official Review · Reviewer_7n3w · 2025-10-30

**Soundness:** 2
**Presentation:** 3
**Contribution:** 3
**Rating:** 4
**Confidence:** 3

**Summary:**

The paper proposes a dual mutual information framework to disentangle embodiement representation and task representation. This disentangled representation learning is then used to edit human demonstration video into a robot demonstration video. Specifically, the two representations are encouraged to be maximally informative of each other within the modality, whereas they are encouraged to have minimal information across the modality. The authors used two different mutual information (MI) estimators. For intramodal MI maximization, InfoNCE estimator is used. For the intermodal MI bottlenecking, CLUB estimator is used. It should be noted that the this cross-embodiment generalization is an emergent result of this disentanglement, not a result of explicit supervised training. The training itself is done in an autoencoding manner without ground-truth cross-embodiment pair.

**Strengths:**

The proposed approach is well motivated and the information-theoretic approach is technically sound. While the idea dual contrastive learning is not new in the so-called multiview representation learning field, its adapation to embodiment-transferrable video robot policy is an important and timely contribution.

**Weaknesses:**

While the methodology itself makes a lot of sense, empirical results are insufficient to support its advantage over baseline methods or applicability to wide variety of videos and embodiments. Specifically, I have two major concerns.

### **[Concern 1]** Metrics used measure video quality, not the transferrability.
I understand that it is incredibly difficult to objectively measure the performance for this kind of problem. That said, the evaluation protocols seems inappropriate or hard to understand.

For instance, the metrics in the Video Fidelity category requires target video sample (PSNR, LPIPS, SSIM) or distribution (FVD). However, there cannot be a ground truth target sample/distribution for this kind of cross-embodiment editing. As such, I believe the evaluation is done in an autoencoding fashion rather than with the unavailable GT cross-embodiment target. If the former, it contradicts the claim that the general purpose VACE performs better in-distribution but does not transfer to cross-embodiment editing. If the latter, it is important to explain how the GT pairs are obtained.

The VBench result is also counterintuitive. If the cross-embodiment transferrability is the key, then the metrics like Subject Consistency (SC) should be higher than VACE. However, the result is the opposite. The proposed method instead excels at more detailed consistency like temporal flickering or motion smoothness. By only looking at these results, I would be more inclined to believe that VACE actually transfers better. It would be helpful if the authors can provide video generated by VACE that corresponds to the one shown in Figure 3.

### **[Concern 2]** Diversity of evaluation settings is limited.
The provided qualitative results (supp. video and figures) are limited in diversity in terms of embodiment and background. These qualitative results are only shown for a single robot and a background. It is also unclear how diverse the quantitative evaludation settings are. Necessary details such as the number of embodiments, backgrounds, and tasks are missing.


As the methodology itself makes sense and the figures/videos looks convincing (although they are limited in diversity), I am willing to adjust my rating to more positive ones if these concerns are addressed.

**Questions:**

Please find the questions in the weakness section.

---

> ### Author Response · Authors · 2025-11-29
> **Response to Reviewer 7n3w**
>
> We sincerely thank the reviewer for the thoughtful evaluation. We appreciate your recognition of our "technically sound" information-theoretic approach and the "well-motivated" problem setting. We are encouraged by your willingness to raise your score and have addressed your concerns regarding evaluation metrics and diversity below.
>
> **[Concern 1] Evaluation Protocols & Ground Truth**
>
> **Our Response:** We appreciate the opportunity to clarify our evaluation methodology.
> 1.  **Pseudo-Ground Truth:** While perfectly paired cross-embodiment data does not exist, the **PH²D dataset** [1] contains "aligned" demonstrations where humans and robots perform identical tasks under similar conditions. We utilize these aligned robot videos as **pseudo-ground truth** for calculating paired metrics. This allows us to quantitatively measure how well the generated video matches the *expected* robot execution, rather than just auto-encoding the input.
> 2.  **VBench & Subject Consistency (SC):** We respectfully point out that VACE’s higher SC score is a "false positive" driven by its tendency to generate artifacts. As VBench SC calculates consistency across frames, VACE achieves a high score by frequently replacing the hand with a black mask-like region that barely changes over time. This is technically "consistent," but semantically incorrect.
>
> **[Concern 2] Diversity of Evaluation Settings**
>
> **Our Response:** Our evaluation covers a broad range of settings.
> * **Embodiments:** We evaluate on **3 embodiments**: the **Unitree H1** humanoid in real and simulation, and Human.
> * **Tasks:** Our evaluation spans multiple distinct manipulation skills, including **Grasping**, **Picking**, and **Pouring**, involving diverse objects (bottles, boxes, cups, etc.).
> * **Environments:** The data includes various tabletop setups with different lighting and clutter.
> * **New Experiments:** Crucially, as detailed in our **General Response**, we have expanded our evaluation with a **downstream policy learning** experiment on the Unitree H1 humanoid in the MuJoCo simulator. This validates our model's ability to generalize to a complex, high-DoF robot within a physics-based environment. By successfully training a control policy on this new embodiment (outperforming the HAT baseline), we demonstrate that our framework effectively scales to diverse and challenging morphologies, going far beyond the single-robot visualization.
>
> We hope these clarifications and our downstream policy validation address your concerns.
>
> [1] RogerQi. PH2D. Hugging Face. https://huggingface.co/datasets/RogerQi/PH2D

---

### Official Review · Reviewer_uB1R · 2025-10-31

**Soundness:** 2
**Presentation:** 2
**Contribution:** 3
**Rating:** 6
**Confidence:** 4

**Summary:**

This work proposes a representation learning framework for disentangling task and embodiment features in egocentric videos. The disentangled representations are then used for cross-embodiment video editing by replacing human hands with robot end-effectors using a video diffusion model. The model is trained with a flow-matching objective and a dual contrastive learning loss that minimizes mutual information between the task and embodiment representations while maximizing intra-space consistency. Experiments on the Physical Human-Humanoid Dataset ($PH^2D$) show that the proposed approach outperforms existing baselines in terms of visual quality and motion consistency metrics.

**Strengths:**

- Cross-embodiment video editing helps to mitigate the embodiment gap between human hands and robot end-effectors in pixel space.
- The proposed dual contrastive objective is effective at learning a disentangled latent space, as evident in Fig.4.
- Learning a disentangled representation for task & embodiment from unpaired cross-embodiment data has the potential to scale up robot learning from human videos.
- Quantitative results in Tab.1 show the benefits over existing baselines in terms of visual quality (FVD, LPIPS, PSNR, SSIM) & motion consistency (TS, MS, OC, TF) metrics.

**Weaknesses:**

- It'd be useful to have a simple baseline that finetunes an existing video editing model (using adapters) on the $PH^2D$ dataset. For example, training the proposed model with only the flow-matching objective from Eq.3. This would help understand if the representation learning framework is indeed effective compared to simpler alternatives like finetuning.
- Tab.2 has an ablation without the dual contrastive loss. Which loss components from Eq.3 are removed in this ablation? It is important to ablate the different loss terms in Eq.3 (i.e., $L_{disentangle}$, $L_{taskcontrast}$, $L_{embcontrast}$) to understand the contribution of each term. Since the disentanglement component is applied only once every ten training steps for stability (L362-363), a systematic ablation would further clarify if this is indeed required.
- As per the CLUB paper, the upper bound on MI is valid if the conditional distribution is known. For unknown conditional distributions, the upper bound holds if a trained variational model can approximate the conditional distribution reasonably well (Theorem 3.2 in the CLUB paper). It'd be useful to clarify this since the proposed work trains a variation model $q_{\phi}$ to approximate the conditional distribution. Also, more details are required about this variation model and how it is trained. Is it a simple VAE or more complex neural network? Is it trained jointly with the cross-embodiment editing framework or separately?
- The text has several mentions of "physically plausible" (L196, L407) and "morphologically accurate/consistent" (L025-026, L083, L407-408, L481). However, there is no evaluation of physical plausibility or morphological consistency. The current evaluation metrics only focus on visual fidelity but do not capture kinematic structure or end-effector physics/dynamics (except in the pixel space). I'd suggest removing these terms from the text since the experiments do not support these claims.
- Visualizations in Fig.3,5,6,7 show that the VACE model often replaces the hand with either a black mask-like region or another hand. It'd be interesting to have some insights into this behavior. Is it because VACE has never seen robot end-effectors or any other reason?

**Questions:**

While learning a task-embodiment disentangled representation space is an elegant idea, I have three concerns (more details in the Weaknesses above):
- A simple finetuning baseline is required to verify if the proposed framework is indeed effective than simpler alternatives.
- Since a dual contrastive objective is a contribution, a system ablation of different loss terms is needed to understand the effectiveness of each term.
- Clarifications regarding the use of "physically plausible" and "morphologically accurate/consistent" terms.

---

> ### Author Response · Authors · 2025-11-29
> **Response to Reviewer uB1R**
>
> We sincerely thank the reviewer for the positive assessment and the constructive suggestions. We are encouraged that you found our disentanglement approach "elegant" and the results promising. We have addressed your concerns regarding baselines, ablation studies, and "physically plausible" below.
>
> **[W1 & Q1] Simple Finetuning Baseline**
>
> **Our Response:** We appreciate this suggestion. The ablation labeled **"w/o Dual Contrastive"** in our submission (Table 2) and the **General Response** is exactly the simple baseline you requested.
> * **Definition:** This model consists of the frozen VACE backbone with our trainable Adapters, trained **only** using the Flow-Matching objective ($\mathcal{L}_{FM}$) from Eq. 3, without any disentanglement or contrastive losses.
> * **Result:** As shown in the qualitative results (Fig. 8 & 9 in the paper and the new videos in the anonymous link), this "simple finetuning" baseline struggles. Without the explicit disentanglement enforced by our dual objective, the model fails to overcome the domain shift, frequently "hallucinating" human hands or producing black artifacts. This confirms that the representation learning framework is essential for effective cross-embodiment editing.
>
> **[W2 & Q2] Detailed Loss Ablation**
>
> **Our Response:** To isolate the contribution of each term in Eq. 3, we have expanded our ablation study as detailed in the **General Response** (Section 2). We trained separate models removing specific components:
> *   **w/o Dual Objective:** Removing the entire Dual Contrastive Objective relies solely on the reconstruction loss ($\mathcal{L}_{FM}$), leading to a lack of overall alignment.
> *   **w/o Disentangle:** The reduced separation between representations in the absence of $\mathcal{L}_{disentangle}$ results from the removal of the mutual information minimization constraint.
> *   **w/o Contrast:** Removing the intra-space contrastive objectives causes structural degradation in their respective domains: omitting $\mathcal{L}_{contrast}$ leads to undefined and overlapping clusters in the Space.
>
> **[W3] CLUB Variational Model Details**
>
> **Our Response:** Thank you for pointing this out. We are happy to clarify the details of the CLUB estimator.
> * **Architecture:** The variational model $q_{\phi}(z_{emb}|z_{task})$ is a Multi-Layer Perceptron (MLP). It consists of three linear layers with GELU activations (and a Tanh output for the log-variance), mapping the condition $z_{task}$ to the mean $\mu$ and log-variance $\log\sigma^2$ of the approximated distribution.
> * **Training:** It is trained jointly with the main framework but with a distinct objective.
>     * **Step A (Variational Update):** We update the parameters $\phi$ of the variational model to *maximize* the log-likelihood of the true pairs $(z_{task}, z_{emb})$, improving the approximation $q_{\phi} \approx p(z_{emb}|z_{task})$. This ensures the upper bound is valid (as per Theorem 3.2 in the CLUB paper).
>     * **Step B (Minimization):** We update the encoders to *minimize* the estimated mutual information calculated using $q_{\phi}$.
>     * This alternating process ensures the bound remains tight while we push the representations to be independent.
>
> **[W4 & Q3] "Physically Plausible" & Downstream Validation**
>
> **Our Response:** We agree that these terms require robotic verification. To substantiate these claims, we conducted the **downstream robotics validation** you suggested as detailed in the **General Response** (Section 1).
> * **Experiment:** We trained a policy using videos generated by our model and compared it to a state-of-the-art baseline (HAT).
> * **Result:** Qualitative and quantitative results confirm that our generated videos are indeed "physically plausible" to serve as effective training data for robotic control.
>
> **[W5] Insight on VACE Failure Modes**
>
> **Our Response:** The behavior of VACE (generating black masks or human hands) stems from its pre-training on large-scale internet video, which is dominated by humans.
> * **Human Hand Hallucination:** When the model receives a masked region and a prompt implying manipulation, its strongest prior is to inpaint a human hand. Without our disentangled embodiment injection, the pre-trained prior overrides the target robot condition.
> * **Black Artifacts:** This typically occurs when there is a conflict between the masked context and the conditioning. The model fails to resolve the "out-of-distribution" request (robot hand in a human scene). Our adapter effectively bridges this gap by aligning the features in the latent space before generation.

---

### Official Review · Reviewer_dyeH · 2025-10-31

**Soundness:** 1
**Presentation:** 2
**Contribution:** 1
**Rating:** 2
**Confidence:** 4

**Summary:**

This paper proposes a generative framework for cross-embodiment video editing, enabling robots to imitate human demonstrations by translating human egocentric videos into robot-execution videos. The method learns disentangled task and embodiment representations via a dual contrastive objective—minimizing mutual information between the two spaces while maximizing intra-space consistency.
By injecting these representations into a frozen video diffusion model, the system can synthesize temporally coherent and morphologically consistent robot videos from human examples without paired data. Experiments show improved visual quality compared to VACE and Phantom baselines.

**Strengths:**

Interesting and timely problem: Tackles the “embodiment gap” in robot learning from human videos — a core challenge in scalable imitation learning.

**Weaknesses:**

- **Evaluation metrics are weak and inconsistent.**: The paper relies primarily on video-generation metrics (FVD, SSIM, LPIPS), which are not well correlated with imitation quality or task performance. In particular, the reported improvements are numerically small and inconsistent across metrics—e.g., LPIPS barely changes, and VACE occasionally performs similarly or better on some sub-metrics. These differences are difficult to interpret and don’t convincingly demonstrate true progress in cross-embodiment understanding.

- **Missing downstream validation.**: Since the stated goal is to enable robot imitation from human video, the key question is whether the generated videos actually improve downstream imitation or policy learning. The paper stops at visual evaluation without measuring how close the generated robot demonstrations are to real robot executions (e.g., comparing to ground-truth robot videos or evaluating imitation performance when trained on generated vs. real data).
This is critical: even if the videos look realistic, they may not encode actionable trajectories or physical consistency necessary for control. A simple behavioral cloning or visual policy learning experiment [1] would make the paper significantly stronger.

- **Quantitative results lack interpretability**: As this is a generative model, it’s understandable that numeric metrics are limited, but the current evaluation doesn’t connect to the end goal. Showing relative fidelity to real robot demonstrations (as an upper bound) would contextualize how useful the generated data actually is.

[1] Kim, Hanjung, et al. "UniSkill: Imitating Human Videos via Cross-Embodiment Skill Representations." arXiv preprint arXiv:2505.08787 (2025).

**Questions:**

Please see weaknesses.

---

> ### Author Response · Authors · 2025-11-29
> **Response to Reviewer dyeH**
>
> We sincerely thank the reviewer for the critical and constructive feedback. We appreciate your recognition of our problem setting as "interesting and timely." We completely agree that visual metrics alone are insufficient for robotics and that downstream validation is essential.
>
> To address your primary concern regarding the lack of robotic verification, we have conducted a comprehensive new set of experiments.
>
> **[W1 & W2] Missing Downstream Validation & Evaluation Metrics**
>
> **Our Response:** We agree that the ultimate test of cross-embodiment video editing is its utility for downstream policy learning. As detailed in our **General Response**, we have implemented a **Visual Behavioral Cloning** pipeline to directly validate the physical plausibility and effectiveness of our generated videos.
>
> * **Experiment Setup:** We utilized the MuJoCo physics simulator with a high-DoF **Unitree H1** humanoid robot performing a "Grasping Pepsi" task.
> * **Method:** We used our framework to translate human demonstration videos into the target robot's visual domain. We then trained an **ACT** [1] policy using these synthesized robot videos.
> * **Baseline:** We compared our method against **HAT** [2], a state-of-the-art method that aligns human and robot data directly for policy learning.
>
> **Results:** As shown in the table below (and Table 3 in the revised manuscript), the policy trained on our generated videos outperforms the HAT baseline.
>
> **Table: Policy Precision on Unitree H1 Grasping Task (Validation Loss $\downarrow$)**
>
> | Method | Total Val Loss | Action L1 Error | EEF Pos Loss |
> | :--- | :---: | :---: | :---: |
> | HAT (Qiu et al., 2025) | 0.2006 | 0.019 | 0.091 |
> | **Ours** | **0.1886** | **0.018** | **0.085** |
>
> These results directly address your concern: our generated videos are not just visually realistic but **physically consistent**, providing better kinematic supervision than implicit alignment methods like HAT. This experiment serves as the "stronger evaluation" you requested, proving that our generated data effectively bridges the embodiment gap for control.
>
> **[W3] Quantitative Results Interpretability**
>
> **Our Response:** We acknowledge that standard video metrics can be opaque regarding task utility. We believe the downstream policy results above provide the interpretability lacking in pure video metrics. To further contextualize the quality, we have provided side-by-side video comparisons of robot rollouts and the generated training data at the anonymous link (**[https://anonymous.4open.science/r/Ego_re-3145](https://anonymous.4open.science/r/Ego_re-3145)**).
>
> We hope these downstream validations and new comparisons effectively address your concerns and demonstrate the practical value of our framework for robot learning.
>
> [1] Zhao, Tony Z., Vikash Kumar, Sergey Levine, and Chelsea Finn. "Learning fine-grained bimanual manipulation with low-cost hardware." arXiv preprint arXiv:2304.13705 (2023).
>
> [2] Qiu, Ri-Zhao, Shiqi Yang, Xuxin Cheng, Chaitanya Chawla, Jialong Li, Tairan He, Ge Yan et al. "Humanoid Policy~ Human Policy." arXiv preprint arXiv:2503.13441 (2025).

---

### Author Response · Authors · 2025-11-29
**General Response (part 2)**

### 2. Comprehensive Ablation Study
To address questions regarding the contribution of each loss component, we performed a systematic ablation study, training separate models: **w/o Disentangle** ($\mathcal{L}\_{disentangle}$), **w/o Intra-space Contrast** ($\mathcal{L}\_{task-contrast}, \mathcal{L}_{emb-contrast}$), and **w/o Dual Objective** (removing both). We provide t-SNE visualizations of the learned latent spaces in the anonymous link.

**Analysis of t-SNE Results:**
*   **w/o Dual Objective:** Removing the entire Dual Contrastive Objective relies solely on the reconstruction loss ($\mathcal{L}_{FM}$), leading to a lack of overall alignment.
*   **w/o Disentangle:** The reduced separation between representations in the absence of $\mathcal{L}_{disentangle}$ results from the removal of the mutual information minimization constraint.
*   **w/o Contrast:** Removing the intra-space contrastive objectives causes structural degradation in their respective domains: omitting $\mathcal{L}_{contrast}$ leads to undefined and overlapping clusters in the space.

### 3. VACE-Finetuned Baseline and Qualitative Results
We explicitly compare our method against a **"VACE-Finetuned" baseline** (equivalent to the **w/o Dual Contrastive** ablation), which trains the adapter using only the flow-matching objective. As shown in the supplementary videos, this baseline fails to bridge the domain shift, frequently producing **hallucinations** (generating human hands instead of robot arms) or **black artifacts**. In contrast, our full model generates morphologically consistent robot end-effectors. We have significantly expanded our qualitative results to include **additional generated videos** for Grasp, Pick, and Pour tasks, alongside the **simulation transfer results** and **robotic policy rollouts**. To ensure transparency, we also provide the specific **embodiment images** used for inference conditioning, **ground truth reference videos**, and direct visual comparisons against both the original **VACE** and the **VACE-Finetuned** baselines.

We hope these additional experiments and clarifications satisfactorily address the reviewers' concerns.

---

### Author Response · Authors · 2025-11-29
**General Response (part 1)**

We sincerely thank the Area Chair and all Reviewers for their insightful and constructive feedback. We are encouraged that reviewers found our problem setting of cross-embodiment video editing "timely and interesting" and our dual contrastive objective "promising."

We understand that the reviewer discussion period has closed. We respectfully ask the Area Chair and reviewers to closely evaluate the new downstream robotics experiments detailed below. Multiple reviewers cited the lack of downstream policy validation as their primary concern. Because training our generative video model on simulation data and performing subsequent policy learning is highly computationally intensive, these experiments took significant time to complete. These new results definitively demonstrate that our generated videos enable successful policy transfer, directly addressing the core critique of the paper.

We have carefully addressed the concerns regarding downstream robotics validation and the necessity of our loss components. We have uploaded a revised manuscript and provided extensive visual results, code, and video comparisons at the anonymous link: **https://anonymous.4open.science/r/Ego_re-3145**

Below, we outline the major updates and additional experiments:

### 1. Downstream Robotics Validation
A primary concern from reviewers was the lack of validation on downstream robot tasks to prove that our generated videos are "physically plausible" for control.

**Clarification:** We originally envisioned our framework primarily as a representation learning tool, where the disentangled *latent representations* would be used to train unified policies in future work. However, acknowledging the reviewers' strong recommendation to validate the immediate utility of the generation quality, we have implemented a direct **Human-to-Robot Video Transfer** pipeline for this rebuttal.

To validate this, we conducted a **Visual Behavioral Cloning** experiment in a physics-based simulation.
*   **Environment:** MuJoCo simulation with a **Unitree H1** humanoid robot.
*   **Task:** "Grasping Pepsi" (a manipulation task).
*   **Policy:** We first train our video editing model on simulation data [1]. And then we train an **ACT** [2] policy with a ResNet backbone.
*   **Comparison:** We compare a policy trained using **HAT** [3] against a policy trained on human data translated into robot videos using our video editing model. All hyperparameters are kept the same: chunk size 100, batch size 16, number of epochs 50 000, and learning rate 1e-4.

**Table 1: Policy Precision on Unitree H1 Grasping Task**
*We report the validation loss (Action L1 and End-Effector Position) on held-out robot trajectories. Lower is better.*

| Method | Total Val Loss ($\downarrow$) | Action L1 Error ($\downarrow$) | EEF Position Loss ($\downarrow$) |
| :--- | :---: | :---: | :---: |
| HAT | 0.2006 | 0.019 | 0.091 |
| **Ours** | **0.1886** | **0.018** | **0.085** |

**Result:** Compared to the **HAT** baseline, which relies on aligning hybrid human-robot data directly, our approach explicitly translates human demonstrations into the target robot's visual domain. This process significantly **reduces the cross-embodiment gap** *before* policy learning, providing a consistent and morphologically accurate signal that benefits training. Consequently, our method achieves consistently lower validation errors across all metrics. We present **video results** of the rollouts (in both egocentric and third-person views) as well as the **intermediate transferred videos** used for training in the anonymous link.

[1] RogerQi. PH2D. Hugging Face. https://huggingface.co/datasets/RogerQi/PH2D

[2] Zhao, Tony Z., Vikash Kumar, Sergey Levine, and Chelsea Finn. "Learning fine-grained bimanual manipulation with low-cost hardware." arXiv preprint arXiv:2304.13705 (2023).

[3] Qiu, Ri-Zhao, Shiqi Yang, Xuxin Cheng, Chaitanya Chawla, Jialong Li, Tairan He, Ge Yan et al. "Humanoid Policy~ Human Policy." arXiv preprint arXiv:2503.13441 (2025).

---

### Author Response · Authors · 2025-12-01
**Summary Comment for the AC (part 2)**

### Summary of reviewer concerns and status

Although the reviewers have not yet responded to the rebuttal, we believe the new experimental results directly satisfy the "necessary conditions" they explicitly set for improving their scores.

| Reviewer | Scores (Soundness / Pres. / Contrib. / Overall) | Downstream Validation | Ablations & Baselines | Method Clarification | Main updates | Follow-up / Current Stance |
| :--- | :--- | :--- | :--- | :--- | :--- | :--- |
| **dyeH** | 1 / 2 / 1, Overall: **2** | **✓** (Major concern: requested policy learning) | – | – | **Conducted Visual BC experiment** on Unitree H1; showed Ours outperforms HAT. Provided rollout videos at the link. | The reviewer's sole reason for rejection was "Missing downstream validation." This is now fully implemented with positive results. We expect this to address the root cause of the negative score. |
| **uB1R** | 2 / 2 / 3, Overall: **6** | **✓** (Asked for "physically plausible" proof) | **✓** (Requested simple finetuning & loss ablation) | **✓** (CLUB details) | **Added "w/o Dual Contrastive" baseline**; added full component ablation with **t-SNE analysis**; clarified CLUB estimator. | **Original Positive Review.** All requests were met. The new robot experiments and t-SNE plots further substantiate the "Soundness" of the method. |
| **7n3w** | 2 / 3 / 3, Overall: **4** | **✓** (Concerned metrics didn't measure transferability) | – | **✓** (Evaluation protocols) | **New policy experiment directly measures transferability**, addressing the "pseudo-ground truth" concern. Clarified VBench scores. | The reviewer explicitly stated a **willingness to adjust the rating** if transferability concerns were addressed. The robot policy results provide the concrete evidence of transferability requested. |
| **8Qp5** | 3 / 3 / 2, Overall: **2** | **✓** (Major concern: "No robot experiment") | **✓** (Requested component-wise ablation) | **✓** (Masks/Inference details) | **Added Robot Policy experiment**; conducted component ablation with **t-SNE**; clarified mask/inference. | Like dyeH, this reviewer's rejection was hinged on the lack of robot tasks and detailed ablations. Both are now included in the revision. |

### Conclusion

The initial reviews were polarized primarily due to a missing piece of evidence: **downstream robotic control validation**. We have successfully closed this gap. The new **Unitree H1 policy learning experiment**—supported by visual rollout results—demonstrates that our method generates data that is physically functional for control. Combined with the requested **t-SNE visualizations** and ablations, we have addressed every major critique raised by the committee. We hope these extensive revisions and new results satisfactorily address the reviewers' concerns.

---

### Author Response · Authors · 2025-12-01
**Summary Comment for the AC (part 1)**

This comment summarizes the state of the reviews and details how the shared concerns—primarily regarding downstream robotics validation and ablation studies—have been comprehensively addressed in the revised manuscript and rebuttal.

Across the four reviews, the substantive critiques converged on two main themes. We have addressed these by conducting significant new experiments, including a full downstream policy learning evaluation and a systematic component-wise ablation study. Extensive visual results and comparisons are provided at: **https://anonymous.4open.science/r/Ego_re-3145**

*   **Downstream Robotics Validation (The Primary Concern):** Raised strongly by **Reviewer dyeH** and **Reviewer 8Qp5**, and noted by **Reviewer uB1R** and **Reviewer 7n3w**. The reviewers argued that visual metrics alone are insufficient to prove the "physical plausibility" required for control.
    *   **Update:** We implemented a **Visual Behavioral Cloning** pipeline in the MuJoCo simulator using a high-DoF Unitree H1 humanoid robot. We trained a policy on videos generated by our model and compared it against the state-of-the-art **HAT** baseline.
    *   **Result:** Our method achieved lower validation loss and action error than the baseline (Table 3 in revision).
    *   **Visual Evidence:** We provided side-by-side videos of the **successful robot rollouts** driven by the policy trained on our data, alongside the generated training videos, at the anonymous link. This definitely proves our generated data is physically plausible for control.

*   **Ablations and Baselines:** **Reviewer uB1R** and **Reviewer 8Qp5** requested a deeper analysis of the loss components and a simple finetuning baseline.
    *   **Update:** We conducted a systematic ablation removing specific loss terms ($\mathcal{L}_{disentangle}$, $\mathcal{L}_{contrast}$, and the full dual objective).
    *   **Result:** We demonstrated that the "simple finetuning" baseline (removing the dual objective) fails to manage domain shift, often hallucinating human hands.
    *   **Visual Analysis:** We included **t-SNE visualizations** (Fig. 4 & Appendix) showing that our full objective creates distinct, well-separated clusters for tasks and embodiments, whereas removing components leads to unstructured, overlapping latent spaces.

---

### Meta-Review · Area_Chair_b9SN · 2026-01-07

**Summary:**

The paper proposes a dual contrastive representation learning framework to address the "embodiment gap" by editing human egocentric videos into robot videos. While the problem is timely, the submission received a mixed-to-negative reception (scores of 2, 6, 4, 2). The primary consensus among reviewers dyeH, 8Qp5, and 7n3w was that the paper fundamentally lacked downstream validation to prove the generated videos were physically plausible for control, rather than just visually consistent. Reviewers also raised significant concerns regarding the ambiguity of the evaluation metrics, the lack of component-wise ablations to justify the complex loss landscapes, and the absence of a simple finetuning baseline.

Although the authors submitted a rebuttal containing a new simulation-based policy learning experiment, the decision remains Reject. The added simulation experiments haven't sufficiently justify the broad "cross-embodiment" claims or address the potential sim-to-real gaps. Overall the paper is not ready for publication at its current form.

**Reviewer Concerns:**

Addressed:
- Lack of Robot Policy Experiment: The authors addressed the request from dyeH and 8Qp5 for "physical plausibility" by implementing a Visual Behavioral Cloning experiment in MuJoCo. They compared their method against the HAT baseline.
- Ablation Studies: The concern raised by uB1R and 8Qp5 regarding the necessity of the loss terms was addressed by adding "w/o Disentangle" and "w/o Contrast" ablations, accompanied by t-SNE visualizations.
- Simple Baseline: The authors added a comparison to a "VACE-Finetuned" baseline (training only the adapter with flow-matching) to address uB1R's request.

Oustanding:
- Metric Inconsistency & Validity: Reviewer 7n3w pointed out that the VBench results were counterintuitive, specifically that Subject Consistency (SC) was lower for the proposed method compared to the VACE baseline. While the authors added a policy experiment, they did not fundamentally resolve why the visual consistency metrics contradict their claims of superior embodiment transfer. If the model fails standard consistency benchmarks, the stability of the RL policy is suspect.
- Limited Diversity: Reviewer 7n3w noted that qualitative results were limited to a single robot and background. The rebuttal added tasks (Grasp, Pick, Pour) but appears to still rely heavily on the single Unitree H1 simulation setup. This does not sufficiently prove the method generalizes across the "internet-scale" diversity implied by the introduction.
- Verification of New Experiments: The policy learning results  are a substantial addition (effectively a new paper's worth of content). As the reviewers did not respond to the rebuttal, there has been no independent verification of the experimental setup, baseline implementation (HAT), or whether the "Visual BC" pipeline is standard or cherry-picked.

**Reviewer Scores:**

The initial reviewer scores are overall negative, with 2 strong rejects. Some of the concerns have been addressed during discussion but i believe the major ones still remain so I don't think the reviewers will significantly increase their ratings.

---

### Decision · Program_Chairs · 2026-01-26

Reject